# Correlation between Physical Performance and Stabilometric Parameters in Older Adults

**DOI:** 10.3390/medicina58091211

**Published:** 2022-09-02

**Authors:** Noé Labata-Lezaun, Vanessa González-Rueda, Jacobo Rodríguez-Sanz, Carlos López-de-Celis, Luis Llurda-Almuzara, Pere Ramón Rodríguez-Rubio, Albert Pérez-Bellmunt

**Affiliations:** 1Department of Basic Sciences, Faculty of Medicine and Health Sciences, Universitat Internacional de Catalunya, 08195 Barcelona, Spain; 2ACTIUM Functional Anatomy Group, Universitat Internacional de Catalunya, 08195 Barcelona, Spain; 3Physiotherapy Department, Faculty of Medicine and Health Sciences, Universitat Internacional de Catalunya, 08195 Barcelona, Spain; 4Fundació Institut Universitari per a la Recerca a l’Atenció Primària de Salut Jordi Gol i Gurina (IDIAPJGol), 08007 Barcelona, Spain

**Keywords:** physical functional performance, elderly, postural balance, accidental falls, stabilometry

## Abstract

*Background and Objectives*: Falls are a common and serious threat to the health and independence of older adults. The decrease in functional capacity during aging means an increased risk of falls. To date, it is not known whether there is a relationship between balance and functional tests. The aim of the study was to evaluate the correlation between eyes-open and eyes-closed static balance with different functional tests. *Materials and Methods*: A correlation study was designed with 52 healthy subjects over 65 years of age. *Results*: Regarding the open eyes stabilometric parameters, significant correlations observed between the surface and the functional tests were weak in all cases. The correlations observed between length and the functional tests performed were moderate, except for that of the Timed Up and Go test (TUG) which was weak. No significant correlation between TUG and surface was found. Regarding the closed eyes stabilometric parameters, statistically significant moderate correlations were found between the surface and the Short Physical Performance Battery (SPPB) and the Five Times Sit to Stand test (5XSST). In the case of the length with eyes closed, a statistically significant moderate correlation (rho = 0.40–0.69) was found with the SPPB and 5XSST variables, and weak correlations with the 4 m Walk Speed test (4WS) and TUG variables. *Conclusions*: There is a mild to moderate correlation between some functional tests and stabilometric parameters in adults over 65 years old.

## 1. Introduction

Falls are a common and serious threat to the health and independence of older adults. According to data from the Centers for Disease Control, approximately one in three older adults suffers a fall each year [1]. In addition, one in five falls results in serious injuries, such as broken bones or head injuries. More than 95% of hip fractures and up to 2.8 million emergency treatments for fall injuries occur each year [2]. In addition, the mortality rates are 20–35% and, of those who survive, between 30% and 45% [3,4] do not recover the functionality of prior to the fracture [5].

Although falls are multifactorial in nature, balance impairment has been identified as one of the main intrinsic risk factors for falls [6]. Physical performance and balance generally decline with ageing, leading to an increased risk of falls for many older adults, especially those who are inactive or insufficiently active [7].

Although there is no universally accepted definition of human balance [8,9], in the clinical setting it can be defined as the inherent ability or postural control of a person to maintain, achieve, or restore a specific state of balance and not to fall [10]. Relevant parameters are the obtained surface area of the ellipse and the length of the stabilogram. However, the cost of stabilometry can be very high and for this reason the assessment of balance in older adults is complicated outside the research setting. Differences in the stabilometry parameters have appeared depending on muscle condition [11] or physical activity level [12], and an alteration in stabilometry has also been seen in different pathological processes [13,14,15].

Physical performance has been defined as an objective measure of body function related to locomotion [16]. In addition, it is closely related to the state of health [17,18,19,20,21], and a significant association between fall risk and muscle weakness has been observed [22]. Results of functional tests such as the Short Physical Performance Battery (SPPB) [19] or the walking speed test [17] have also been shown to be associated with the risk of falls [23], disability [21], and even mortality [19,20]. These functional tests are frequently useful in the clinical field and experts recommend that all older adults be screened for physical performance in primary care to detect those at risk for frailty and/or sarcopenia [16].

In this way, the literature shows that interventions focused on improving functional capacity have also proven to be effective in reducing falls and balance [24]. Therefore, it is reasonable to think that there is a direct relationship between physical performance and balance. However, currently, no study has been found that correlates those variables. If there is a correlation between the two, using functional tests instead of a stabilometric platform may be a more affordable option for the clinical setting.

The aim of this study was to evaluate the correlation between eyes-open (OE) and eyes-closed (CE) static balance with different functional tests in people over 65 years old.

The hypothesis of this study was that there is a statistically significant correlation between OE and CE static balance and different functional tests in people over 65 years old.

## 2. Materials and Methods

### 2.1. Study Design

A cross-sectional observational correlational study was performed.

### 2.2. Sample Size Calculation

The GRANMO v 7.12 program was used to calculate the sample size. A correlation coefficient analysis was performed with an alpha risk of 0.05, bilateral contrast, beta risk of 0.20, a moderate Pearson correlation coefficient estimate of 0.4 and a loss forecast of 10%, and a necessary sample of 52 subjects was obtained.

### 2.3. Participants

Fifty-two volunteers were recruited through the Universitat Internacional de Catalunya and the Associació de Gent Gran Casal Anna Murià (Tarrasa). At the university, students and faculty staff were verbally informed of the characteristics of the study and the inclusion and exclusion criteria, and our contact telephone number was left in case volunteers knew of any other person who might meet the requirements. At the senior center, a voluntary meeting was held with the users to explain the study and to obtain the contact details of persons who may be interested in participating. The participant selection process was carried out by telephone call. The measurements were carried out between October 2020 and November 2021.

Inclusion criteria were: (a) people over 65 years of age, and (b) being able to perform all the assessment tests of the study. Exclusion criteria were: (a) inability to stand or ambulate in an unassisted manner, (b) previous bone fracture in the previous 6 months, (c) uncontrolled symptomatic cardiovascular or respiratory disease, (d) current cancer under treatment, and (e) inability to understand the information provided by the assessors.

### 2.4. Outcomes

The primary variable of balance measured with stabilometry and the secondary variables of functionality were recorded through the SPPB, the 4 m walk test (4WT), the Five Times Sit to Stand test (5XSST) and the Timed Up and Go test (TUG).

Balance was assessed by means of the stabilometric variables of surface area of the elipse and length of the stabilogram. The surface of the ellipse comprises 95% of all the measured points of the center of pressures. It is measured in square millimeters. A larger surface area of the ellipse implies a lower capacity to maintain equilibrium at the center of pressure. The length of the stabilogram comprised 100% of the points recorded. It is measured in millimeters, and assesses the accuracy of the fine postural system in maintaining balance. A longer stabilogram length indicates a greater involvement of the fine control system in rebalancing. Both parameters were measured with eyes open and with eyes closed. The main difference between both measurements is that with the eyes open, the visual system is mainly involved, whereas, with the eyes closed, the vestibular system, the proprioceptive system, and the plantar receptors become more important [25,26]. Measurements were performed using the Satel 40 Hz stabilometric force platform (model PF2002; SATEL SARL, 6 rue du Limousin-31,700 Blagnac, France). This is a portable platform used in clinical and research settings, which has proven to be a valid and reliable tool for measuring balance in the standing position [27]. Measurements were performed in a standardized position, with the feet at a 30° angle, and the heels 2 cm apart. Subjects were positioned facing a white wall with a red plumb line 90 cm from the platform to ensure that the subject is centered. Subjects were asked to keep their arms at the side of the body and to relax as much as possible, without clenching the jaw (Figure 1). [27]. The recording time is 51.2 s [28], determined by the platform’s ability to collect 40 data per second with a conversion card (2048 data captured per minute).

The SPPB test battery is widely used in primary care and research. It consists of 3 tests: a balance test with feet together, in semi-tandem and tandem positions for 10 s with eyes open; a walking speed test over a distance of 4 m; and a test of sitting down and getting up from a chair 5 times. Each score in the three tests has a value of 0–4, which was summed to give a maximum score of 12 points. Previous studies have established that a score equal or less than 8 points is associated with an increased risk of future negative events, such as falls, hospitalizations, sarcopenia, or frailty [19]. Test-retest reliability has been shown to be good to excellent (ICC between 0.83 and 0.92 for measurements taken one week apart), and inter-rater reliability is excellent (ICC 0.91) among acutely admitted elderly patients [23].

The 5XSST shows the time in seconds it takes a person to sit down and stand up five times from a chair with a backrest and without arm assistance. Although it is included in the SPPB battery, its score has individual value on its own. The test was performed twice and the one with the shorter time was chosen. Previous studies have established that a time greater than 15 s is associated with an increased risk of future negative events, such as frailty, sarcopenia, falls, or hospitalization [16].

The 4WS shows the time in seconds it takes a person to travel 4 m at normal speed. Although it is included in the SPPB battery, its score has individual value. The test was performed twice and the one with the shortest time was chosen. Its reliability has been previously studied (ICC = 0.96, 95%CI = 0.94–0.98; SEM = 0.01) [29]. Previous studies have established that a speed lower than 0.8 m/s (3.2 s for the 4 m) would be related to a higher risk of suffering future negative events, such as frailty, sarcopenia, falls, or hospitalization [16,30].

The TUG is a test commonly used in both consultation and research. It reflects the time in seconds it takes the person to get up from the chair, with the help of the arms, walk 3 m, turn around an obstacle, return to the chair, and sit down again. The test was performed twice and the one with the shorter time was chosen. Previous studies have established that a time above 20 s indicates the person is at risk for future negative events [16]. In addition, its reliability has been previously studied (ICC = 0.98, 95%CI = 0.93–1.00; SEM = 0.7).

### 2.5. Procedure

After contacting the subjects, it was verified that they met the inclusion/exclusion criteria. Before starting the study, all participants were asked to sign the informed consent form. To begin, personal data, height and weight, and dominance were recorded. Next, the functional tests (SPPB, TUG, 5XSST, and 4WS) were performed. Finally, stabilometry was performed in a quiet room. All measurements were performed standing, barefoot, and without socks. Measurements were repeated if the patient coughed, sneezed, yawned, turned his or her head, or performed a maximal inhalation [27]. Both OE and CE measurements were repeated twice. From the two measurements, the values having the best score were chosen [25].

### 2.6. Statistical Analysis

Statistical analysis was carried out with the Jamovi v.1.6.23 [Computer Software]. Descriptive statistics were calculated for all variables. Quantitative variables were expressed as mean and standard deviation, and qualitative variables as number and percentage. The Kolmogorov Smirnov test was used to test the normal distribution of the variables. Correlation analysis was performed by calculating Pearson’s correlation coefficient or Spearman’s rank correlation coefficient, depending on the distribution of the variable evaluated. The significance level was set at 0.05 with a 95% confidence interval. The following intervals were used to interpret the correlation coefficient: 0–0.10 insignificant correlation, 0.11–0.39 weak correlation, 0.40–0.69 moderate correlation, 0.70–0.89 strong correlation, and 0.90–1.00 very strong correlation [31].

## 3. Results

Between October 2020 and November 2021, 52 subjects over 65 years of age (31 men and 21 women) who met all eligibility criteria and agreed to participate were recruited. The demographic characteristics of the sample are summarized in Table 1. No adverse effects, side effects, or losses were recorded in the study.

After performing the Kolmogorov Smirnov test, all dependent variables in this study followed a non-normal distribution (Kolmogorov Smirnov test *p* < 0.05) with the exception of the gait speed variable (Kolmogorov Smirnov test *p* > 0.05). Table 2 records the descriptive values of the functional tests and stabilometry.

The results of the correlation analysis are shown in Table 3. In the tests with OE, we observed a statistically significant correlation in all variables for both surface and length, with the exception of the surface of the TUG test.

The correlation observed for the surface with OE was, in all cases, weak. However, the correlation values that we observed for the length in the test performed with OE were moderate, except for that of the TUG test, which was weak.

Regarding the tests with CE, we observed a statistically significant moderate correlation in the SPPB and the 5XSST for the surface with CE.

In the case of the length with CE, we observed a statistically significant moderate correlation in the SPPB and 5XSST variables and a weak correlation in the 4WS and TUG variables.

## 4. Discussion

The aim of the present study was to evaluate the relationship between static balance with OE and CE with different functional tests in people over 65 years of age.

Regarding the descriptive values, all the means of the values of the functional tests were above the cut-off points indicating risk of suffering future negative events [23]. With regard to the results of stabilometry, to date no studies have been found that assess balance with this device. The study by Rodríguez-Rubio et al. [27] assessed 42 healthy adults aged 18–65 years. Their results were lower for both surface and length in OE and CE. The results of the present study are surely different due to the age of the volunteers. This would suggest that stabilometric parameters change with increasing age (more surface and length). However, it is unclear whether these changes are due to the lower level of physical activity of older adults (as suggested in previous investigations [12]) or the age of this sample. In the neurological field, alteration of stabilometric parameters has been described in Alzheimer’s disease [15], multiple sclerosis, and fragility patients [14,32].

The interpretation of the correlations observed between stabilometry and the different variables indicates that a better score (values closer to 12) on the SPPB implies better stabilometric parameters. With regard to the 5XSST variable, we observed that the longer the time required to complete the test, the worse the values for the stabilometric parameters. In the 4WS variable, the shorter the time required to complete the test, the better the stabilometric parameters, and finally, in the TUG test, the shorter the time required to complete the test, the better the stabilometric parameters. Although there was a significant correlation in all these variables, the correlation varied between mild and moderate, so the results should be interpreted with caution.

An earlier systematic review with meta-analysis concluded that exercise interventions are beneficial in improving physical function by increasing muscle strength, gait speed, mobility, balance, and physical performance [26]. Indirectly, we observed this close relationship between balance and the functional capacities of the subjects. Another systematic review with meta-analysis by Gine et al. [33] examined the effects of exercise on physical function in older adults, and showed that exercise was effective in improving normal gait speed (MD = 0.07 m/s), fast gait speed (MD = 0.08 m/s), and SPPB score (MD = 2.18). According to different official organizations, multicomponent programs focused on strength training are the best strategy to improve the physical performance of older adults [34,35]. In the future, it would be interesting to assess whether a multicomponent training program is able to improve both physical performance and stabilometric parameters.

Among all the functional tests, the SPPB had the highest level of correlation with the stabilometric parameters. These results are consistent, since one-third of the total score of this test is obtained in the static balance test. In this sense, it is possible that the correlations obtained are not strong because the functional tests not only evaluate balance, but also strength and gait speed.

Based on the results obtained, we consider that both functional tests and stabilometric parameters have their clinical utility, but should not be interchanged. Functional tests would provide more general information on health status, and would be more useful in the screening process, whereas stabilometric parameters are much more specific and should be used mainly in people with some type of balance or postural control impairment.

### Study Limitations

The main limitation of this study is that it was only performed in healthy older adults, and we do not know if these results would follow a similar correlation in older adults having different pathologies. Another limitation we observed is that these functional tests can only be used in people with certain functionality, and are totally dependent on the patient’s ability to ambulate. Finally, the fact that the level of physical activity performed by these individuals was not recorded may be a limitation to be taken into account in future projects, as there is an association between the physical activity level and the stabilometric parameters

## 5. Conclusions

There is a mild to moderate correlation between different functional tests and stabilometry in adults over 65 years old.

## Figures and Tables

**Figure 1 medicina-58-01211-f001:**
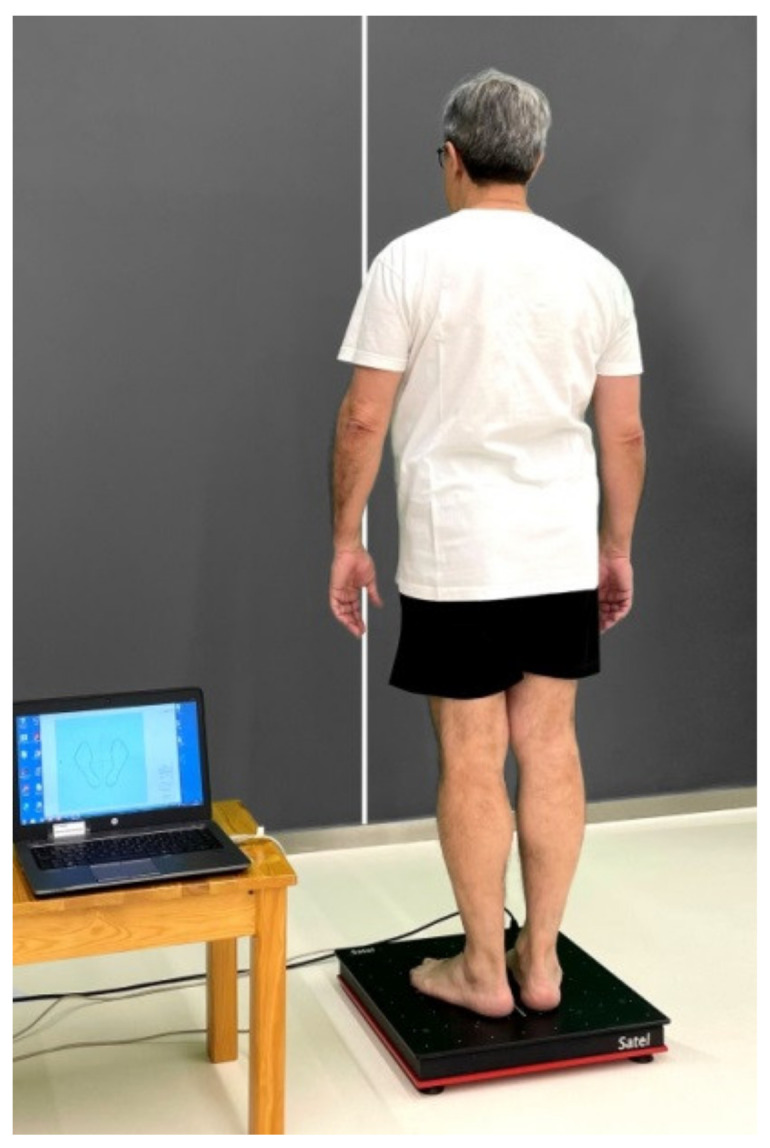
Stabilometric measurement.

**Table 1 medicina-58-01211-t001:** Anthropometric data of the sample.

Variable	Mean ± SD n°
Sex (Men/Female)	52 (31/21) ^+^
Dominance Lower Limb (Right/Left)	52 (50/2) ^+^
Dominance Upper Limb (Right/Left)	52 (50/2) ^+^
Age (years)	73.7 ± 7.44
Height (cms)	159 ± 10.3
Weight (kgs)	67.4 (20.4) *
BMI (Kg/m^2^)	28.3 ± 4.12

SD, standard deviation; ^+^, number of cases; *, Median (Interquartile range).

**Table 2 medicina-58-01211-t002:** Functional test and estabilometric parameters of the sample.

	SPPB (0–12)	5XSST (sg)	4WS (m/sg)	TUG (sg)	OE Surface (cm^2^)	OE Length (cm)	CE Surface (cm^2^)	CE Length (cm)
Mean	11.2	12.1	1.07	9.04	215	540	306	748
Median	12.0	10.3	1.07	8.50	167	494	224	662
Standard Deviation	1.50	6.34	0.260	2.59	163	202	224	333
Interquartile Range	1.00	2.50	0.233	2.73	124	277	262	360

SPPB, Short Physical Performance Battery; 5XSST, Five Times Sit to Stand test; 4WS, 4 m Walk Speed test; TUG, Timed Up and Go test; OE, Open Eyes; CE, Closed Eyes.

**Table 3 medicina-58-01211-t003:** Correlation analysis.

		SPPB	5XSST	WS	TUG
OE Surface	Spearman’s rho	−0.384	0.278	−0.310	0.239
	*p*-value	**0.005**	**0.046**	**0.025**	0.092
OE Length	Spearman’s rho	−0.491	0.478	−0.448	0.324
	*p*-value	**<0.001**	**<0.001**	**<0.001**	**0.020**
CE Surface	Spearman’s rho	−0.462	0.502	−0.265	0.256
	*p*-value	**<0.001**	**<0.001**	0.058	0.070
CE Length	Spearman’s rho	−0.542	0.534	−0.362	0.292
	*p*-value	**<0.001**	**<0.001**	**0.008**	**0.037**

SPPB, Short Physical Performance Battery; 5XSST, Five Times Sit to Stand test; 4WS, 4 m Walk Speed test; TUG, Timed Up and Go test; OE, Open Eyes; CE, Closed Eyes.

## Data Availability

Not applicable.

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
