# Peer review of "Correlation between Physical Performance and Stabilometric Parameters in Older Adults"

_medicina, 2022, doi:10.3390/medicina58091211_

Round 1

Reviewer 1 Report

Thank you so much for allowing me to review this paper.

This research paper indicates the correlation between physical performance and stabilometric parameters in older adults. Therefore few studies explain the physical performance and stabilometric parameters in older adults. 

Abstract 

Q1 There is too much numerical information in the abstract; readers may understand the background and how the results reflect future needs.

Introduction

Q2. Discuss the background of physical performance and stabilometric parameters and their relevance and significance to older adults. 

Methods

Q3. Explain more details about the recruitment process 

Q4. Line 160-161: How is the sample size significant? Suggested the calculation of the sample size in the paper.

Q5 Men and females (N) are not equal; will this affect the results?

Results

Q6 line 165-166, a non-normal distribution (p< .05), can explain more the p-value and the behind these results. 

Q7 Line 213-217 can be the sub paragraph" limitations"

Discussion 

Q8 Based on the results, any new insights?

Conclusion

Q9. Any suggestions for the intervention in the future?

Reviewer 2 Report

1. TUG in Abstract: Please use full forms when using a term for the first time.

2.  Please explain well the differences between OE/CE Surface and Length tests. How were they performed? How long were the tests? These assessments are the basis of all analyses and conclusions, so need to be clear.

3. It is unclear how the results obtained from this study may suggest that balance worsens with age. There were no control group for age, either older or younger.

4. It is unclear how the results imply better SPPB scores mean better balance. There is no data for worse/lower scores with physical function variables to draw this conclusion.

5. Is there any data on daily exercise/activity levels of the participants? 

6. Overall, the study lacks control groups, whether it is people with different pathologies affecting their mobilities, or with younger, healthier people with better mobilities. The authors need to either address these limitations better and explain their conclusions accordingly, or they need variable participant groups and repeat the same tests on them. The authors should also rewrite the discussion to address this serious shortcoming of the study design.

Round 2

Reviewer 2 Report

Reviewer 2 - Review Report (Round 1)

Comments and Suggestions for Authors

First of all, thank you very much for taking the time to read and review our work.

1. TUG in Abstract: Please use full forms when using a term for the first time.

Thank you very much for your comment. We have revised and corrected the abbreviations both in the abstract and in the rest of the manuscript.

2.  Please explain well the differences between OE/CE Surface and Length tests. How were they performed? How long were the tests? These assessments are the basis of all analyses and conclusions, so need to be clear.

Thank you for your input. We have added more information in the “outcomes” section (lines 112-120). Regarding the duration of the measurements, and the procedural aspect of the tests, we have kept the information in the procedure section (lines 156-168). If you do not agree with this distribution, we can move the information to the variables section.

Please put all background information specific to any one test/assessment in the same section, be it methods/outcomes/procedure. If you divide different aspects of the same test in different sections it will be difficult for the readers.

3. It is unclear how the results obtained from this study may suggest that balance worsens with age. There were no control group for age, either older or younger.

Thank you very much for your contribution. Indeed, the present study is an observational study. The conclusion obtained in Line 261 refers to the fact that, in this cohort of elderly people, stabilometric values were found to be higher than the reference values in middle-aged adults in the study by Rodríguez-Rubio et al. In this sense, we consider it pertinent to justify the hypothesis that perhaps the greatest difference between the two study populations was age, and that it was therefore a factor influencing balance. In any case, if you prefer, we can replace the word "equilibrium" with "stabilometric parameters" to adjust somewhat more closely to reality. Of course, this is not one of the main conclusions of the study, but it opens doors to future lines of research such as the one you propose.

Can you use the data available from that previous study as a control in this study? If not, then explain in depth how you infer the data obtained from the present study based on the previous study. Replacing the word "equilibrium" with "stabilometric parameters" will be better.

4. It is unclear how the results imply better SPPB scores mean better balance. There is no data for worse/lower scores with physical function variables to draw this conclusion.

Thank you again for your comment. In our opinion, the study design (correlation study) does allow us to make this claim. Indeed, the people who have obtained better physical performance values have obtained better values in the stabilometric parameters, with a higher or lower level of correlation depending on the type of functional test. Once again, if required, we can replace the word "equilibrium" with "stabilometric parameters" to adjust somewhat more closely to reality.

I think replace the word "equilibrium" with "stabilometric parameters" will be better.

5. Is there any data on daily exercise/activity levels of the participants?

Thank you very much for your recommendation. Indeed, it would be very interesting to know what level of physical activity the participants had. Unfortunately, we do not have that information. We will take it into account for future studies in this line of research.

Please add a study limitation section to discuss all the limitations of the study so that it helps the readers to acknowledge them and come to their own conclusions.

6. Overall, the study lacks control groups, whether it is people with different pathologies affecting their mobilities, or with younger, healthier people with better mobilities. The authors need to either address these limitations better and explain their conclusions accordingly, or they need variable participant groups and repeat the same tests on them. The authors should also rewrite the discussion to address this serious shortcoming of the study design.

Thank you very much for your contribution. Indeed, no control group has been performed, as this is an observational study of correlations, not a clinical trial. In this sense, the aim of the study was only to establish the level of correlation between the stabilometric parameters and the functional tests. At the same time, in the discussion section, the descriptive values of the sample have been compared with those obtained in other studies in other types of younger populations. Regarding pathologies, as we commented in the limitations section, it is important to be clear that the results of the present study should not be extrapolated to multipathological populations.

Finally, the sample size was based on an analysis without subgroups, so it is impossible to divide the subgroups without losing statistical power. We hope that our answer has convinced you, and we remain at your complete disposal.

Please make changes according the suggestions.
